# Neuroinflammation and Brain Health Risks in Veterans Exposed to Burn Pit Toxins

**DOI:** 10.3390/ijms25189759

**Published:** 2024-09-10

**Authors:** Athena W. Brooks, Brian J. Sandri, Joshua P. Nixon, Timothy R. Nurkiewicz, Paul Barach, Janeen H. Trembley, Tammy A. Butterick

**Affiliations:** 1Minneapolis Veterans Affairs Health Care System, Minneapolis, MN 55417, USA; brooksa@umn.edu (A.W.B.); sand0449@umn.edu (B.J.S.); nixon049@umn.edu (J.P.N.); trem0005@umn.edu (J.H.T.); 2Medical School, University of Minnesota, Minneapolis, MN 55455, USA; 3Department of Surgery, University of Minnesota, Minneapolis, MN 55455, USA; 4Department of Food Science and Nutrition, University of Minnesota, St. Paul, MN 55108, USA; 5Department of Physiology, Pharmacology, and Toxicology, West Virginia University, Morgantown, WV 26506, USA; tnurkiewicz@hsc.wvu.edu; 6Center for Inhalation Toxicology, West Virginia University, Morgantown, WV 26506, USA; 7The Department of Safety and Quality Science in the College of Population Health, Thomas Jefferson University, Philadelphia, PA 19144, USA; paul.barach@jefferson.edu; 8Department of Laboratory Medicine and Pathology, University of Minnesota, Minneapolis, MN 55455, USA; 9Masonic Cancer Center, University of Minnesota, Minneapolis, MN 55455, USA; 10Department of Neuroscience, University of Minnesota, Minneapolis, MN 55455, USA

**Keywords:** air pollution, neuroinflammation, neurodegenerative disease, burn pit toxins, Veterans, brain–lung axis, mental health disorders, occupational health

## Abstract

Military burn pits, used for waste disposal in combat zones, involve the open-air burning of waste materials, including plastics, metals, chemicals, and medical waste. The pits release a complex mixture of occupational toxic substances, including particulate matter (PM), volatile organic compounds (VOCs), heavy metals, dioxins, and polycyclic aromatic hydrocarbons (PAHs). Air pollution significantly impacts brain health through mechanisms involving neuroinflammation. Pollutants penetrate the respiratory system, enter the bloodstream, and cross the blood–brain barrier (BBB), triggering inflammatory responses in the central nervous system (CNS). Chronic environmental exposures result in sustained inflammation, oxidative stress, and neuronal damage, contributing to neurodegenerative diseases and cognitive impairment. Veterans exposed to burn pit toxins are particularly at risk, reporting higher rates of respiratory issues, neurological conditions, cognitive impairments, and mental health disorders. Studies demonstrate that Veterans exposed to these toxins have higher rates of neuroinflammatory markers, accelerated cognitive decline, and increased risks of neurodegenerative diseases. This narrative review synthesizes the research linking airborne pollutants such as PM, VOCs, and heavy metals to neuroinflammatory processes and cognitive effects. There is a need for targeted interventions to mitigate the harmful and escalating effects of environmental air pollution exposures on the CNS, improving public health outcomes for vulnerable populations, especially for Veterans exposed to military burn pit toxins.

## 1. Introduction

The scientific community has dedicated research to determining the systemic effects of environmental toxins from air pollution on human health and wellness. Air pollution has traditionally been associated with increased risks for pulmonary and cardiovascular disease [1,2], and since 2002, the growing relationship between central nervous system (CNS) diseases and air pollution has been explored [3]. Research on this growing relationship has shown an increased prevalence of neurodegenerative diseases, such as Alzheimer’s and Parkinson’s, in both animal test subjects and humans after repeated exposures to air pollutants. Inflammation and oxidative stress are determined to be common mechanisms for air pollution-induced chronic pulmonary and cardiovascular diseases. Similarly, previous research has shown that air pollution-induced neuroinflammation seems to be the culprit of neurodegenerative pathologies [4]. In addition to neuroinflammation and neurodegeneration, air pollution has also been shown to increase the incidence of depression and other mental health diagnoses [5,6].

Military burn pits contain toxins present in air pollution, but research on the effects of burn pit toxins on Veterans is not as extensive as the effects of airborne toxins on urban populations. Burn pit emissions often comprise a more complex and hazardous toxin mixture than urban air pollution derived from military waste that consists of plastic, wood, metals, and miscellaneous combustive materials [7]. The major differences between military waste and civilian waste include more packaging, construction materials, food waste, Meals Ready to Eat (MRE) waste, and plastic water bottles, which are burned and cause burn pit toxins to be extremely volatile [7]. In addition to particulate matter (PM), burn pit toxins also contain poly-aromatic hydrocarbons (PAHs), toxic metals, and volatile organic compounds (VOCs). Previous research on the effects of air pollution on human brain health has the potential to help assess the increased risk military personnel incur after they are exposed to burn pits during their deployment. We found one review analyzing burn pit toxins and neurological pathologies, and one review analyzing burn pit toxins and mental illnesses [8,9]. While the reviews are informative, there is a need to more deeply understand the concepts of neuroinflammation and specific pathologic mechanisms.

In this integrative literature review, we gather and synthesize both empirical and theoretical evidence relevant to how air pollutants affect and penetrate the blood–brain barrier (BBB) leading to neuroinflammation, neurodegeneration, cognitive impairment, and mental illnesses. Exposure to air pollutants is not limited to urban pollution and can be found in Veterans deployed in the Middle East and Southeast Asia exposed to burn pits. We relate air pollution-induced neurologic and psychological changes with burn pit-induced changes.

## 2. Methods

The goal of this review is to connect concepts of air pollution, neuroinflammation, neurodegenerative diseases, and mental health and relate them to burn pit toxins and the escalating public health risks that Veterans face.

### 2.1. Search Strategy and Selection Criteria

We searched the PubMed (including MEDLINE) and Scopus databases in July 2024 for all published literature from 2008 to July 2024, with supplementary searches of these databases and Google Scholar occurring up to November 2023. The databases selected were comprehensive and included only papers published in English. The search query terms included “PM AND neurodegeneration”, “burn pits AND neurodegeneration”, “PM AND mental health”, “burn-pits AND mental health”, and “neuroinflammation AND PM AND neurodegeneration” (Table 1).

We focused on the published literature that targeted the relationships between pollution and neuroinflammation, heavy metals and neurodegeneration, and post-9/11 Veterans and neurodegeneration with mental health. Experimental studies including mice and humans, retrospective cohort studies, and reviews were included in the search. The references of the eligible studies were manually checked (snowballing), to identify additional relevant studies that were missed in the database searches [10]. The initial selection for inclusion was based on a single reviewer assessment of the titles and abstracts that were seen in the first ten articles from the Google Scholar and PubMed search (AB) (Table 1). A full–text copy of each eligible study was examined, and if the article was a review, the snowballing method was used (AB). Many search results had repetitive information and/or studies, or non-related studies, and these studies were excluded. Specific papers by Yoshida et al., Romaní-Aumedes, et al., Ellisen, and Pérez-Sisqués, et al. were relied on to highlight interesting correlations between cigarette-smoke-induced inflammation and neurodegeneration [11,12,13,14]. These papers did not show up in the literature review search.

### 2.2. Inconsistencies and Biases in the Literature

Results from human-based studies on burn pit exposure and retrospective studies were not present due to the lack of research efforts and robust epidemiological data. This study relied on human and rodent studies analyzing the relationships between PM, VOC/PAH, and heavy metals and neurodegenerative and mental health illnesses. This was implemented to draw parallels between urban air pollution and burn pit toxins because these pollutants exist in both environments. Additionally, we relied on findings based on acute exposures of these toxins. Although this review uses literature non-specific to burn pit toxins, we believe that the similarities of the toxins to urban pollution help to underscore the translation of potential health effects to Veterans. 

## 3. The Lung–Brain Axis

Air pollutants are one of the most prevalent sources of environmentally induced inflammation and oxidative stress that cause CNS diseases [4]. A novel study by Calderon-Garciduenas examined the causal relationships between pollution and neurodegeneration by investigating feral dogs exposed to high levels of pollution [3]. Enhanced oxidative damage, the presence of amyloid plaques, tissue damage, and accumulated metals, along with similar pathologies of patients with Alzheimer’s and Parkinson’s diseases, were observed in dogs exposed to toxins. More recent studies have confirmed this relationship and demonstrated that BBB damage and endothelial cell activation is also present, potentially impairing signal conduction and the progressive loss of neurological functions [15,16,17]. 

The BBB is a specialized system of microvascular endothelial cells that prevents toxic substances from entering the brain tissue while providing the brain with nutrients [18]. Environmental toxins can affect the inflammation and permeability of the blood–brain barrier (BBB). PM has been found in both brain parenchyma and blood capillaries in humans, suggesting that toxic pollution compounds can penetrate the BBB. Endothelial cell damage in the BBB can be caused by aluminum nanoparticles, which increase intracellular adhesion molecules (ICAMs) and vascular cell adhesion molecules (VCAMs), both of which induce vascular inflammation and endothelial leakiness [19,20,21]. Rat studies have shown that there is also an increase in ROS and cytokine production in the brain parenchyma in response to PM, which can contribute to CNS pathology, including neurodegenerative diseases [4]. Additionally, all drug availability may be decreased due to the PM-induced upregulation of efflux transporters (P-glycoprotein and multidrug-resistance-associated protein-2) and the downregulation of tight junctions at the BBB [22]. Overall, the air pollution PM can lead to a damaged endothelial layer in the BBB, the production of cytokines, and ROS, which signal changes in transporter expression and function, and the upregulation of VCAM/ICAM in the cerebral vasculature. These BBB changes give toxins the opportunity to cause neuroinflammation, neurodegeneration, and mental illnesses (Figure 1).

Due to the increased permeability and damage of the BBB, PM can directly interact with the brain parenchyma and activate microglia [22]. Along with pollutants directly interacting with the brain parenchyma through a damaged leaky BBB endothelium, cytokines from peripheral inflammatory responses may also activate microglia [22]. Increases in CD68 (expressed in the microglia), CD163 (associated with endothelial cell adhesion and oxidative damage protection), and HLA-DR (a marker for effector T cells) positive cells were also found in brain tissue analyses from individuals residing in highly polluted areas, indicating microglia activation, in addition to the elevation of pro-inflammatory markers, such as IL-1β, TNFα, and COX-2 [18,19]. Microglia have pattern recognition receptors that identify large pathogen-associated molecular patterns, including differing charge and protein aggregates. The inflammatory response caused by PM and ultrafine particulate matter (UFPM) in respiratory epithelial cells can introduce surface charge to the microglia, which can make PM and UFPM proinflammatory stimuli when they reach the brain [22]. Microglia are activated through exposure to pollutants, similar to the effects of cytokines, neuronal death, and endogenous disease proteins. Microglia have been shown to respond to particulate matter (PM) in a study using cultures exposed to diesel exhaust particles [3,23,24]. These cultures also demonstrated that microglia mediated neuronal damage and that “microglia-derived [reactive oxygen species (ROS)] are key for [diesel exhaust particle (DEP)-induced] dopaminergic neurotoxicity”, along with neurotoxic titanium nanoparticles with ROS, which are neurotoxic [23]. Manganese, a component of industrial-derived air pollution, can also activate microglia and amplify dopaminergic neurotoxicity in vitro, adding to the ROS build up and neuroinflammation associated with air pollution [25] (Figure 1).

## 4. Toxin-Induced Neuroinflammation

The brain capillaries recognize air pollution and promote an inflammatory response by regulating the BBB and distributing ROS, cytokines, and PM to the brain parenchyma, which contributes to CNS pathology. Lung pathologies are due mainly to inflammatory responses rather than other direct causes of pathologic symptoms. An example of this phenomenon was demonstrated by Yoshida et al., where Rtp801 (also known as REDD1) knock-out mice exposed to tobacco smoke were protected from emphysema, in contrast to wild-type mice who died from tobacco smoke due to emphysema [11]. Rtp801 is a stress-response protein that is a major mediator of emphysema induced by smoking, and its expression is triggered by cigarette smoke, which leads to oxidative stress and alveolar inflammation, leading to cell death [11,13]. This mechanism is proven due to the activation of NF-κB, leading to neutrophil, macrophage, cytokine storms, and lymphocyte recruitment. Additionally, Rtp801 inhibits mTORC1, which leads to the downregulation of HIF-1 and ROS, which causes apoptosis of the epithelial cells in the alveoli [13].

Interestingly, Rtp801 also plays a role in mediating human and mouse Alzheimer’s disease (AD) and juvenile parkinsonism [12,14]. Human postmortem hippocampal samples from AD patients had significantly increased amounts of Rtp801 expression, which was also seen in transgenic mice that had AD pathology, specifically 5xFAD and rTg4510 mice [14]. Rtp801 was elevated in Parkin double-knockout (KO) mice brains and fibroblasts from juvenile parkinsonism patients, which is caused by PARK2 gene mutations affecting parkin solubility and E3 ligase impairment, which can lead to slow, progressive neuronal degeneration and cell death [12]. Human postmortem Parkinson’s disease (PD) brains demonstrated a 75% increase in the expression of Rtp801 in fibroblasts compared to unaffected individuals [26]. The suspected mechanism was posited that Parkin can protect neural cell death initiated by Rtp801 overexpression through mediating Rtp801 degradation, which was not seen in Parkin knockout mice. This evidence suggests that air pollution-related CNS pathology, such as sporadic AD and PD, could mainly be due to the inflammatory response systemically, and more specifically at the BBB (Figure 1).

Extensive evidence points to at-risk populations having an increased likelihood of developing neuroinflammation after exposures to air pollution, such as in Veterans exposed to open-air burn pits. Burn pit emissions have a mixture of carcinogens and neurotoxins, including PM less than 2.5 μm, polyaromatic hydrocarbons (PAHs), and volatile organic compounds (VOCs). The seminal Framingham Heart Study demonstrated that reduced pulmonary function was seen in environmental PM2.5 exposures that were up to 30 times lower than exposures experienced in post-9/11 deployment areas [27]. Military personnel from repeated deployments have demonstrated altered and inflammatory markers associated with chronic diseases, including lung cancer, bronchitis, and pulmonary fibrosis. We have previously used mice models to confirm that inhaled PM causes a systemic inflammatory response, and the resulting chronic inflammation induces dysfunction in the pulmonary system. Our mechanistic focus includes the onset of inflammation due to NF-κB activation and CK2 downregulation, resulting in oxidative, mitochondrial, and ER stress, leading to chronic lung injury. Given the systemic inflammation that is induced by PM, PAHs, and VOCs, these Veterans are at higher risk for developing BBB damage caused by systemic inflammation induced by PM, PAHs, and VOCs, and could also be the culprit for the observed increased incidence of sporadic Alzheimer’s disease and related dementias (ADRD) (Figure 2).

## 5. Toxin-Induced Neurodegeneration/Cognitive Impairment

Air pollutants have been shown to cause an increase in ADRD, memory decline, and cognitive deficits [22,28,29]. Specifically, the heavy metals found in polluted air are some of the most associated toxins with neurodegenerative diseases. These heavy metals include lead, arsenic, and manganese [30]. Inorganic arsenic, the active component of chromated copper arsenate, uncouples oxidative phosphorylation and impairs glucose metabolism [31]. When comparing the control and exposure of 3xTg mice to inorganic arsenic from gestation to 6 months of age, a decline was observed in complex I and an oxidative state in the hippocampus, along with an increased antioxidant response in the cortex [32]. There was also an increased amount of phosphorylated tau in the frontal cortex and hippocampus. This experiment suggests that mitochondrial dysfunction may be an important triggering factor for chronic arsenic-induced exacerbation of AD-like pathology [32,33].

Epidemiological studies in humans have demonstrated that lead exposure can lead to early indications of Alzheimer’s disease, including the accumulation of heavy metals and the inflammation of neurons [34]. Lead exposure can mechanistically induce neuronal apoptosis and microglia activation. Notably, the pro-inflammatory proteins induced by lead are largely attributed to neurodegeneration in Parkinson’s disease, in which abnormal forms of α-Synuclein can trigger selective and progressive neuronal death through mitochondrial impairment, lysosomal dysfunction, and the alteration of calcium homeostasis [35].

In addition to heavy metals, polyaromatic carbons (PACs) and volatile organic compounds (VOCs) are also responsible for neurodegenerative diseases. We have previously validated in rats that whole-body inhalation of carbon black, a surrogate for PM2.5, can accumulate in lung tissue and increase biomarkers of inflammation [36]. Additional preliminary studies using a mixture of carbon black and the VOC/PAH chemical naphthalene for whole-body inhalation exposures have demonstrated an increase in lung biomarkers for inflammation and stress associated with lung injury. The brain–lung axis is crucial for mediating these effects. Lung exposure to airborne toxins initiates inflammatory responses impacting brain health, highlighting how lung inflammation can lead to neuroinflammation and cognitive impairments (Figure 2).

Additionally, a recent study reported that an increase in PM2.5-induced inflammatory markers can directly impact higher incidences of EGFR-driven lung cancer [37]. They found that IL1-beta can trigger the expansion of pre-existing mutant lung cells that cause an increased incidence of adenocarcinoma [37]. This information suggests that pollutant-induced neuroinflammation could not only cause neurodegeneration but also brain cancers. Notably, research at UC Davis has reported that Veterans exposed to burn pits in the post-9/11 era have an increased incidence of brain cancer [9,38]. The current literature regarding the relationships between brain tumors and PM is inconclusive beyond associations without direct causal evidence [8,39,40]. Focusing on PM-derived neuroinflammation on brain tissue is a potential direction to determine the presence of causal relationships [8]. Moreover, burn pits contain higher amounts of PM, acrolein, combustible material, and burned plastic waste that contain VOC/PAHs, which have been linked to worsened mental health. We would like to emphasize that these are speculative statements. Prior research points in the direction of direct causal experimental evidence of burn pit-specific toxins affecting Veterans’ brain health through neuroinflammation.

## 6. The Impacts of Air Pollution on Veteran’s Mental Wellbeing

Veterans are at significant risk of developing mental illnesses, such as depression, due to their chronic exposures to toxins present in burn pits. There is both clinical and pathological evidence that environmental toxins can cause cognitive declines and increase the incidence of depression and anxiety [41]. In rat and mouse models, PM leads to adverse neuroinflammatory responses and autoimmune responses, which have the potential to worsen psychiatric conditions, specifically through the Nrf2/NLRP3 pathway [6]. Enhanced oxidative stress after PM2.5 exposure can activate astrocytes via the Nrf2 pathway, as Nrf knockout mice exposed to PM demonstrate significantly upregulated pro-inflammatory cytokine expression and NLRP3 inflammasome activities, which has previously been reported to worsen depression [6,42,43]. Other authors suggest that another possible pathway by which pollutants trigger psychiatric conditions is via increased glucocorticoid activity and stress hormone cortisol concentrations [5].

There is direct pathological evidence that air pollution can lead to depression beyond the documented increased incidence of depression with high PM exposure [29,44]. Male mice exposed to PM2.5 produced elevated hippocampal pro-inflammatory cytokine expression, along with decreased dendritic branching in the hippocampal CA3 regions in comparison to mice exposed to filtered air [29]. Additionally, the mRNA for TNF-alpha and IL1-beta were elevated in PM2.5 mice as compared to filtered-air mice. This finding suggests causal relationships between air pollution and depression because IL1-beta has been associated with “reduced hippocampal plasticity and related changes in depressive-like behaviors” [44].

The PM-induced neuroinflammation effects are not limited to neurodegenerative disorders, but also impact an increased incidence of psychiatric disorders and cognitive impairment. A meta-analysis on PM2.5 and psychiatric illnesses showed that patients exposed to both long-term and short-term PM2.5 were associated with higher rates of depression, anxiety, and suicide [45]. Additionally, a retrospective cohort study conducted in 2021 to examine the relationships between psychotic/mood disorders and air pollution demonstrated that people with higher residential air pollution used mental healthcare services more frequently [46]. However, the authors noted that there may be a confounding variable of ethnic disparities to healthcare, since Black British people are more likely to be forcibly hospitalized than White British people due to the Mental Health Act 1983 [46]. Similar phenomena have been observed in the U.S. and globally, where urban populations are more susceptible to pollution-related illnesses, with confounding variables being the low-income populations who cannot afford healthcare [47,48]. Veteran-specific airborne exposures to heavy metals can also cause ROS, which are contributors to depression, anxiety, and schizophrenia [8,49,50,51,52]. There is limited research analyzing the relationship between burn pits and mental health, and much potential to explore it further [8].

Environments with high amounts of toxins can elevate the risk of certain populations for neurodegenerative diseases, along with pollution-induced pulmonary and cardiovascular diseases. It is important to note that these environments are not limited to urban, highly populated areas, but are also present in burn pit toxins for Veterans (Figure 2). 

## 7. Risks for Veterans and Other Communities

Exposure to air pollution has been associated with depression and emotional disorders, even in non-chronic situations, such as observed in World Trade Center rescue and recovery workers [53]. With our current knowledge of how air pollutants can cause neurodegeneration, neuroinflammation, and decline in mental well-being, we recognize the gaps in healthcare for populations exposed to these pollutants, including for Veterans who served near burn-pits, urban populations, and in farming communities.

Veterans face significant health risks from burn pit toxins, often leading to severe neuroinflammatory responses, accelerating cognitive decline, and increasing neurodegenerative disease risk. Studies show higher rates of neuroinflammatory markers, cognitive impairments, and mental health disorders in Veterans exposed to burn pits [8,54]. The evidence underscores the critical links between airborne pollution and neuroinflammation, significantly impacting brain health, cognition, and mental health. Moreover, burn-pit toxins include metals and PM2.5, which are akin to the contents of air pollutants [55]. The evidence highlighted in this review demonstrate clearly that metals and PM2.5 causing neurodegenerative diseases and mental health issues. This makes screening for sporadic Alzheimer’s and Parkinson’s diseases and depression in burn pit-exposed Veterans important. 

Urban communities are often prioritized when thinking about populations at risk for pollution-induced diseases. However, populations affected by air pollution are not limited to these areas, and many rural communities are vulnerable to these diseases as well. Rural counties have a greater density of water pollution sources and greater agriculture-related air pollution, which are associated with increased cancer mortality rates [56]. Additionally, rural coal mining areas can expose populations to increased cancer and respiratory disease mortality rates [56]. Further research regarding the specific pollutants and at-risk populations can lead to better clinical outcomes.

## 8. Limitations

Our study has inherent limitations. First, our review only covered two databases, PubMed and Google Scholar, so other literature may have been missed. Second, the literature reviewed was limited to papers in the English language that were published from 2008 until July 2024. Third, only the titles and abstracts were used to determine whether a study or review would be relevant for this review. If the title and abstract of the paper was deemed fitting for this review, then the full text was read and analyzed to include in this literature review. Therefore, if relevant information was not found in the title and abstract but was present in the main text, the literature was excluded. Fourth, a narrative review is more descriptive with a broader scope in comparison to more robust systematic reviews, which have well-defined research questions. Furthermore, the connections we suggest between burn pit toxins and neurodegenerative and mental illnesses are limited to speculations based on the past literature. No direct causal relationships between burn pit toxins and neurodegenerative and mental health illnesses have been reported in this review, which opens up future possibilities for experimental studies to make this connection. 

Despite these limitations of our review, narrative reviews are highly useful to educators and researchers. While a systematic review often focuses on a narrow question in a specific context, with a prespecified method to synthesize findings from similar studies, a narrative review includes a wide variety of studies and provides an overall summary with interpretation and critique [57]. Authors can provide insights on advancing the field, new theories, or current evidence viewed from different or unusual perspectives [58]. In this aspect, we highlight the under-researched field of burn pit toxin effects on the brain health of Veterans. A future, more comprehensive systematic review may be necessary to provide deeper understanding of the relationships among burn pits, neuroinflammation, neurodegeneration, and mental illnesses.

## 9. Conclusions

The toxins within burn pits are capable of causing endothelial damage to Veterans, which increases the permeability of their BBB, leading to increases in ROS, cytokine production, and neuroinflammation. There is strong evidence that toxin-induced neuroinflammation is a source of neurodegenerative disease, as seen in cigarette-smoke-induced Rtp801-mediated inflammation causing AD and PD pathologies [11,12,13,14,59]. Cigarette smoke can exacerbate burn pit toxin injuries, and a high percentage of Veterans smoke cigarettes regularly [60]. Smoking may be a confounding variable in assessing the health of Veterans and may lead them to a high risk of developing AD and PD via Rtp801 increase, in addition to neurodegenerative risks caused by burn pit toxins. Heavy metals can also increase tau phosphorylation, leading to AD, along with neuronal apoptosis and microglia, leading to PD through dysfunctional alpha-synuclein. Retrospective and pathologic studies have shown residents in highly polluted environments to have an increased prevalence of mental health illnesses. The toxin-induced neuroinflammation effects are not only limited to neurodegeneration but also include increased incidences of mental health illnesses. Neuroinflammation is linked to neuropathology linked to depression, such as NLRP3 inflammasome activities and decreased hippocampal plasticity. Populations that live in areas with PM2.5 levels, along with Veterans exposed to burn pit toxins, may be more likely to develop depression, anxiety, and schizophrenia. Prolonged burn pit animal models to assess the brain–lung axis are needed to better replicate the pathological tissue responses on organs. Overall, this review adds to previous reviews by Penuelas et al. and Hoisington et al. on how toxin-induced neuroinflammation can contribute to AD, PD, and mental illnesses [8,9]. These reviews, along with ours, showcase a need for more extensive experimental research to explore the effects of burn pit toxins on Veterans’ brain health in order to lessen health disparities and give proper health care to affected Veterans [61].

## Figures and Tables

**Figure 1 ijms-25-09759-f001:**
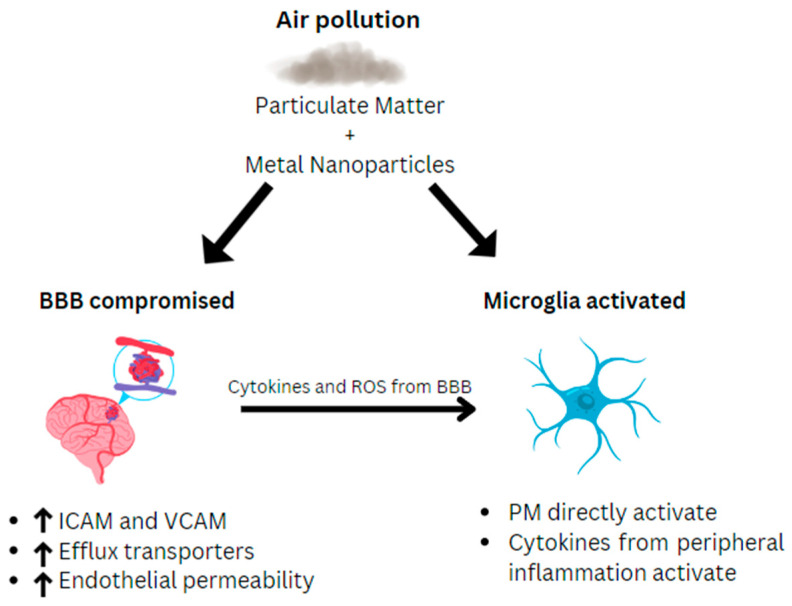
The relationships among air pollutants, the blood–brain barrier (BBB), and microglia. Particulate matter and metal nanoparticles are able to increase the permeability of the BBB by increasing intracellular adhesion molecules (ICAMs), vascular cellular adhesion molecules (VCAMs), and efflux transporters, which allow for peripheral inflammatory cytokines and reactive oxygen species (ROS) caused by the pollutants to enter the brain parenchyma and thus activate the microglia.

**Figure 2 ijms-25-09759-f002:**
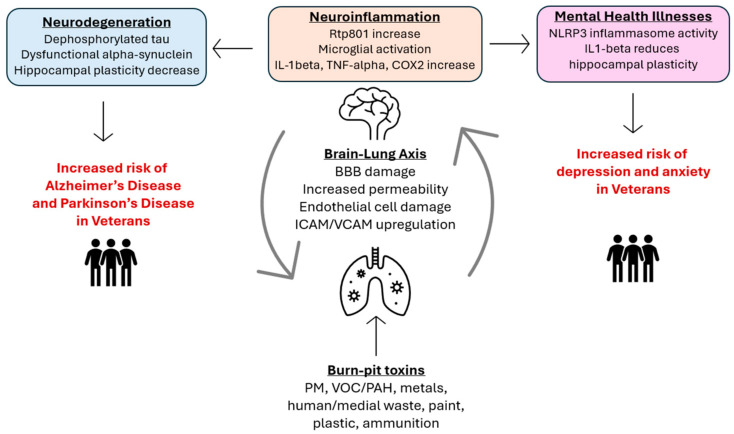
An overview of the lung–brain axis and diseases which are affected by it. Toxins inhaled cause increased cytokines and reactive oxygen species (ROS), along with endothelial cell damage, intracellular adhesion molecule/vascular cell adhesion molecule (ICAM/VCAM) upregulation, and increased permeability, all of which contribute to blood–brain barrier (BBB) damage. At the same time, these toxins that cross the BBB cause neuroinflammation through a number of pathways, including Rtp801 increase, inflammatory cytokine increase, and microglial activation. These neuroinflammatory mechanisms lead to neurodegeneration, depression, and anxiety.

**Table 1 ijms-25-09759-t001:** Inclusion and exclusion criteria table. We used “PM AND neurodegeneration”, “burn pits AND neurodegeneration”, “PM AND mental health”, “burn pits AND mental health”, and “neuroinflammation AND PM AND neurodegeneration” as the search query terms. The table shows the parameters for inclusion and exclusion criteria we used to choose the literature for this review.

Category	Inclusion Criteria	Exclusion Criteria
Study types	Animal studies Human studies Systematic reviews	Abstracts Case series/reports Editorials Letters to Editors
Year published	Literature published between 2008–July 2024	Literature published before 2008 and after July 2024
Language	Literature published in English	Literature published in non-English languages
Search results	Top 10 literature results from PubMed and Google Scholar databases using search query terms	Literature that were not in the top 10 search results
Content	Relating to connecting neuroinflammation with neurodegeneration and/or mental health Air pollution and/or burn pit toxins’ effects on neurodegeneration and mental health illnesses via neuroinflammation	Repetitive Content Content not relating to connecting neuroinflammation with neurodegeneration and mental health

## Data Availability

No new data were created or analyzed in this study. Data sharing is not applicable to this article.

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
