# Peer review of "Neuroinflammation and Brain Health Risks in Veterans Exposed to Burn Pit Toxins"

_ijms, 2024, doi:10.3390/ijms25189759_

Round 1

Reviewer 1 Report

Comments and Suggestions for Authors

The manuscript by Brooks and colleagues addresses a relevant topic, given the unfortunate cumulative high number of past and ongoing military conflicts worldwide and the need to clarify the link between airborne pollutants and neuroinflammation, neurodegeneration, and cognitive dysfunction. The study's relevance is further underlined by the limited number of works directly addressing these issues in the context of burn pits.

Despite its current flaws, which I detail below, I believe this paper fits the scope of the IJMS. However, I believe it must undergo major revisions before it can be considered for publication, mainly in order to add some information on the molecular alterations and interactions leading to neurotoxicity following occupational exposure to airborne toxicants.

General remarks

1.     According to the journal’s instructions for Authors, “Acronyms/Abbreviations/Initialisms should be defined the first time they appear in each of three sections: the abstract; the main text; the first figure or table. When defined for the first time, the acronym/abbreviation/initialism should be added in parentheses after the written-out form.” This is not always observed (e.g., “CNS” in the Introduction section – line 43). Please revise this issue throughout the manuscript.

2.     Please adapt all bibliographical references to the recommended reference style.

Section 2.1

3.     I suggest that the inclusion and exclusion criteria be stated more objectively (for instance, in the form of a table).

Section 3

4.     The role of microglia in air pollution-induced neurotoxicity and neuroinflammation (and the underlying molecular changes/interactions) could be further explored within the text and in the form of a figure.

5.     BBB alterations subsequent to the exposure to air pollutants (and the underlying molecular changes/interactions) could be further explored within the text and in the form of a figure.

Section 7

6.     Line 303: There is a period lacking between “issues” and “This”.

Comments on the Quality of English Language

Appropriate English language has been used throughout the text. There are only some written-out forms missing for some abbreviations.

Author Response

We sincerely thank the reviewers for their time, effort, and valuable feedback. We appreciate the opportunity to revise our manuscript and have addressed all the concerns and suggestions as detailed below.

Comment 1:  "According to the journal’s instructions for Authors, 'Acronyms/Abbreviations/Initialisms should be defined the first time they appear in each of three sections: the abstract; the main text; the first figure or table. When defined for the first time, the acronym/abbreviation/initialism should be added in parentheses after the written-out form.' This is not always observed (e.g., “CNS” in the Introduction section – line 43). Please revise this issue throughout the manuscript."

Response: Thank you for pointing this out. We have carefully reviewed the manuscript to ensure that all abbreviations are defined the first time they appear in the abstract, main text, and each figure/table caption.

Comment 2: Please adapt all bibliographical references to the recommended reference style.

Response: We have revised the references according to the journal's guidelines. We have followed MDPI’s guidelines for bibliographical references and have used this format: “In the text, reference numbers should be placed in square brackets [ ] and placed before the punctuation; for example [1], [1–3] 176 or [1,3]. For embedded citations in the text with pagination, use both parentheses and brackets to indicate the reference number 177 and page numbers; for example [5] (p. 10), or [6] (pp. 101–105).” Please let us know if further adjustments are needed, and we will be happy to comply.

Comment 3: "I suggest that the inclusion and exclusion criteria be stated more objectively (for instance, in the form of a table)."

Response: We agree that presenting the inclusion and exclusion criteria in a table format enhances clarity and objectivity. A table has been added in line 114 with a corresponding caption to reflect this change.

Comment 4: "The role of microglia in air pollution-induced neurotoxicity and neuroinflammation (and the underlying molecular changes/interactions) could be further explored within the text and in the form of a figure."

Response: We appreciate this suggestion and have incorporated a new figure (Figure 1) on line 164 that illustrates the underlying molecular changes in pollution-induced neurotoxicity and neuroinflammation.

Comment 5: "BBB alterations subsequent to the exposure to air pollutants (and the underlying molecular changes/interactions) could be further explored within the text and in the form of a figure."

Response: This change has been reflected in the text (lines 146-148) and in Figure 1, which now includes a depiction of BBB alterations due to air pollutants.

Comment 6: "Line 303: There is a period lacking between 'issues' and 'This'."

Response: Thank you for catching this. The error has been corrected in line 327.

We have made every effort to address the concerns of both reviewers comprehensively. We hope that these revisions meet the expectations of the reviewers and the journal Editor, and that the revised manuscript will be found suitable for publication.

Thank you for your consideration.

Reviewer 2 Report

Comments and Suggestions for Authors

The manuscript entitled “Neuroinflammation and Brain Health Risks in Veterans Exposed to Burn Pit Toxins” is generally well written and interesting. It covers the topic of  the systemic effects of environmental toxins from air pollution on human health with emphasis on burn-pit toxins and pollution-related CNS pathologies in veterans. This review synthesizes recent research linking airborne pollutants such as PM, VOCs, and heavy metals to neuroinflammatory processes and cognitive effects. In my opinion the part of the manuscript sacrificed on possible molecular mechanisms linking air pollution, neuroinflammation and neurodegenerative pathologies is clear and well documented. However, the part on burn-pit pollution in veterans and incidence of CNS diseases is more of speculative nature. As explained by the authors mostly due to lack of research in this area.

Major issues:

In order to justify the title of this review the authors should add a paragraph analyzing the incidence of CNS diseases related to air pollution in veterans versus general population. Also, it would be interesting to discuss whether burn pit pollution differs (contents) from “regular air pollution” and its health effects.

Paragraph 3 should be rearranged to show, firstly, how the pollutants get into the blood and cross the blood-brain barrier (BBB).

The description/definition of “CD-68, CD-163, and HLA-DR positive cells” should be provided.

The “limitations” section is to some extent irrelevant and does not address the true limitations or pitfalls of the study/manuscript but rather describe the issue of narrative reviews.

Minor issue:

All over the manuscript the naming and symbols of the molecules should be uniform and according to the official symbols, i.e. CD-68 should be CD68.

Paragraph 3 – shouldn’t it be “The Lung-Brain Axis”?

Author Response

We sincerely thank the reviewers for their time, effort, and valuable feedback. We appreciate the opportunity to revise our manuscript and have addressed all the concerns and suggestions as detailed below.

Comment 1: In order to justify the title of this review the authors should add a paragraph analyzing the incidence of CNS diseases related to air pollution in veterans versus general population. Also, it would be interesting to discuss whether burn pit pollution differs (contents) from “regular air pollution” and its health effects.

Response: We have added a new paragraph that discusses the incidence of CNS diseases related to air pollution in veterans compared to the general population, as well as the differences between burn pit pollution and typical air pollution and their respective health effects. It can be found in lines 57-60 and 263-268.

Comment 2: Paragraph 3 should be rearranged to show, firstly, how the pollutants get into the blood and cross the blood-brain barrier (BBB).

Response: The third section of the manuscript has been rearranged to first describe how pollutants compromise the BBB, followed by an explanation of how this aggravates neuroinflammation and neurotoxicity (lines 129-263).

Comment 3: The description/definition of “CD-68, CD-163, and HLA-DR positive cells” should be provided.

Response: The definitions of CD68, CD163, and HLA-DR positive cells have been provided in lines 149-151.

Comment 4: The “limitations” section is to some extent irrelevant and does not address the true limitations or pitfalls of the study/manuscript but rather describe the issue of narrative reviews.

Response: We agree and have revised the limitations section to more clearly address the specific limitations of the study. We now acknowledge that the claims about the relationship between burn pit toxins and brain health are largely speculative and emphasize the need for further experimental studies (lines 348-353, 359-361).

Comment 5: All over the manuscript the naming and symbols of the molecules should be uniform and according to the official symbols, i.e. CD-68 should be CD68.

Response: We have standardized the naming and symbols of molecules and cytokines throughout the manuscript to adhere to the official nomenclature.

Comment 6: Paragraph 3 – shouldn’t it be “The Lung-Brain Axis”?

Response: We have updated the manuscript to reflect this change, replacing "Brain-Lung axis" with "Lung-Brain axis" throughout the text.

We have made every effort to address the concerns of both reviewers comprehensively. We hope that these revisions meet the expectations of the reviewers and the journal Editor, and that the revised manuscript will be found suitable for publication.

Thank you for your consideration.

Round 2

Reviewer 1 Report

Comments and Suggestions for Authors

In general, the Authors have addressed my concerns. I would just like to highlight the following:

1.     I believe that, though citations throughout the text have been correctly adapted, references in the final list of bibliographical references do not exactly follow the recommended reference style.

2.     Although I expected a greater effective increase in information regarding the role of microglia and BBB alterations in pollution-induced neurotoxicity and neuroinflammation (specifically regarding the underlying molecular changes/interactions), the Authors have addressed this concern mainly by creating Figure 1.

Author Response

Comment 1: I believe that, though citations throughout the text have been correctly adapted, references in the final list of bibliographical references do not exactly follow the recommended reference style.

We have attempted to follow this guideline that is in the template of MDPI for the bibliography section:

"1. Author 1, A.B.; Author 2, C.D. Title of the article. Abbreviated Journal Name Year, Volume, page range. 180"

If this is not what you were envisioning, the IJMS editorial office has kindly offered to help edit this on their end. Thank you for bringing this to our attention.

Comment 2: Although I expected a greater effective increase in information regarding the role of microglia and BBB alterations in pollution-induced neurotoxicity and neuroinflammation (specifically regarding the underlying molecular changes/interactions), the Authors have addressed this concern mainly by creating Figure 1.

We have added additional explanation within the main body to help explain Figure 1 in lines 152-156 and 138-139. 

Reviewer 2 Report

Comments and Suggestions for Authors

I am satisfied with the corrections introduced in the manuscript.

Author Response

Comments: I am satisfied with the corrections introduced in the manuscript.

Response: We thank you for your time and consideration.